# The Influence of Edaphic and Climatic Factors on the Morphophysiological Behavior of Young Argan Plants Cultivated in Orchards: A Comparative Analysis of Three Regions in Southwest Morocco

**DOI:** 10.3390/plants14010126

**Published:** 2025-01-04

**Authors:** Fatima Ezzahra Tiouidji, Assma Oumasst, Salma Tabi, Naima Chabbi, Abdelaziz Mimouni, Meriyem Koufan, Naima Ait Aabd, Abdelghani Tahiri, Youssef Karra, Jamal Hallam, Redouan Qessaoui, Rachid Bouharroud, Fouad Elame, Nadya Wahid, Ahmed Wifaya

**Affiliations:** 1Integrated Crop Production Research Unit, Regional Center of Agricultural Research of Agadir, National Institute of Agricultural Research, Avenue Ennasr, BP 415 Rabat Principale, Rabat 10090, Morocco; assma.oumasst@edu.uiz.ac.ma (A.O.); salma.tabi@usms.ma (S.T.); n.chabbi.ced@uca.ac.ma (N.C.); abdelaziz.mimouni@inra.ma (A.M.); meriyem.koufan@inra.ma (M.K.); naima.aitaabd@inra.ma (N.A.A.); abdelghani.tahiri@inra.ma (A.T.); youssef.karra@inra.ma (Y.K.); jamal.hallam@inra.ma (J.H.); redouan.qessaoui@inra.ma (R.Q.); rachid.bouharroud@inra.ma (R.B.); fouad.elame@inra.ma (F.E.); 2Laboratory of Environmental, Ecological and Agro-Industrial Engineering (LGEEAI), Faculty of Science and Technology of Beni Mellal, Sultane Molay Slimane University, Beni Mellal 23000, Morocco; n.wahid@usms.ma; 3Laboratory of Biotechnology and Valorization of Natural Resources (LBVRN), Faculty of Sciences Ibn Zohr University, Agadir 80000, Morocco; 4Laboratory of Agrobiotechnology and Bioengineering, Department of Biology, Faculty of Science and Technology-Gueliz, Cadi Ayyad University, Marrakesh 40000, Morocco

**Keywords:** *Argania spinosa*, young plants, plant morphophysiology, climatic parameters, soil parameters

## Abstract

*Argania spinosa* (L.) Skeels is a unique endemic species in Morocco, renowned for its ecological characteristics and socio-economic importance. In Morocco, recent years have seen an exacerbation of the harmful effects of climate change, leading to an alarming decline in the natural regeneration of this species in its original habitats. It seems that the only viable solution lies in the domestication of this genetic heritage. This study marks the first in-depth investigation of the impact of various climatic and edaphic factors on the morphological and physiological traits *of Argania spinosa* young plants, assessed in six separate orchards and observed over four seasons (March 2022 (Winter), June 2022 (Summer), November 2022 (Autumn), and March 2023 (Winter)). A climatic assessment was carried out at each site, including measurements of rainfall, maximum and minimum temperatures, mean temperature, air temperature, and wind speed. The soil was analyzed for the pH, electrical conductivity (EC), water content, limestone (CaCO_3_), Kjeldahl nitrogen (N), available phosphorus (P_2_O_5_), organic matter (OM), and carbon/nitrogen ratio (C/N). To gain a better understanding of the morphophysiological characteristics of young argan seedlings, we carried out various observations, such as measuring the height and diameter of aerial parts, and the water content of leaves (WCL) and branches (WCB), quantifying chlorophyll (mg/m^2^) and leaf area. The results revealed a significant impact of edaphic and climatic factors on the morphophysiological parameters of young argan trees. Results revealed significant correlations of young argan plants between edaphic and climatic factors and morphophysiological parameters. The Tamjloujt site, characterized by protective vegetation cover, showed optimal growth conditions with the highest leaf and branch water content (46.89 ± 4.06% and 37.76 ± 3.51%, respectively), maximum height growth (91.33 ± 28.68 mm), trunk diameter (24.85 ± 3.78 mm), and leaf surface area (69.33 ± 19.28 mm^2^) during Summer 2022. The Saharan zone of Laqsabi exhibited peak chlorophyll concentrations (506.9 ± 92.25 mg/m^2^) during Autumn 2022, due to high temperatures. The mountainous environment of Imoulass negatively impacted plant growth (mean height: 52.61 ± 12.37 mm; diameter: 6.46 ± 1.57 mm) due to harsh climatic and edaphic conditions. This research provides vital knowledge regarding the environmental factors influencing the establishment of young argan plants within the Argan Biosphere Reserve. This contributes to the development of more effective domestication strategies and the restoration of agroecosystems. The aim is to use this knowledge to promote the rehabilitation and sustainability of argan agroecosystems.

## 1. Introduction

Climate change significantly affects global ecosystems, particularly in arid and semi-arid regions, where its implications on the growth of trees and plants are profound. The argan tree, a species endemic to Morocco, spans approximately 800,000 hectares across the Western center of Morocco [1]. Despite exhibiting drought resistance, the tree’s density and surface coverage have suffered declines over time. Studies by Lybbert et al. and Guinda et al. [2,3] reported a drastic contraction in the argan tree’s distribution. Almost half of the argan forest has disappeared, with the average tree density in the remaining areas plummeting from 100 trees per hectare to fewer than 30. El Ghazali et al. [4] noted that the Moroccan Ministry of Agriculture recorded an annual reduction averaging 600 hectares in both the density of argan trees and the area they cover. In response, governmental efforts and organizational support have led to the establishment of 10,000 hectares of argan plantations in desertification-prone regions. Numerous studies have explored the influence of climatic conditions on argan growth, examining both in vitro and greenhouse settings, as well as real-world field conditions. For instance, researchers have utilized controlled environments to investigate the growth and development of argan seedlings [5]. Other studies assessed the agro-morphological variability of argan trees under varying environmental circumstances within Morocco, highlighting its significance for selection purposes [6]. Additionally, field research has been conducted to better understand the growth dynamics and adaptation of argan trees to diverse climatic conditions [6,7,8,9]. Notably, Díaz-Barradas et al. [7] identified several traits crucial for understanding how argan trees respond to their environments, notably the interplay of climate variation and seasonality under natural conditions.

Previous studies demonstrate that arid and semi-arid regions with limited precipitation and high evaporation rates tend to experience salt accumulation in the root zones, leading to reduced water percolation [10,11]. The application of saline irrigation water affects soil conditions, which is a vital but finite natural resource [12], contributing to salinization issues. Morocco is recognized as particularly vulnerable to climate change, with water scarcity being a pressing concern, as indicated by the Intergovernmental Panel on Climate Change (IPCC) [13].

A comprehensive understanding of climatic conditions and irrigation practices is essential to grasp the evolution of soil physicochemical properties and their effects on plant growth. Variations in the physicochemical properties of soil and vegetation can be significant, often correlating with lower nitrification rates, adjusted pH, and organic carbon levels [14,15,16]. Several investigations have focused on how irrigation practices affect soil organic carbon content. Olsson et al. and Gebeyehu and Soromessa [17,18] discovered that irrigation fosters increased input of crop residues, thereby greatly affecting soil organic carbon levels. However, Al-Zu’bi [19] points out that the presence of salinity in irrigated fields can diminish soil permeability and productivity potential.

Both drought and salinity pose challenges to the functionality of agroecosystems in arid and semi-arid environments [20,21], affecting various biochemical and physiological processes within plants, such as the hydrological partitioning in root zones [22], overall growth [23,24,25], productivity metrics, chlorophyll levels, and photosynthetic activity [26,27,28]. The argan tree has evolved multiple morphological and physiological adaptations to combat such stresses, such as leaf drop in severe drought conditions [29,30], stomatal regulation, and decreased leaf water potential, along with lower relative water content [31]. Furthermore, the plant often curtails its metabolic processes during drought periods [32] while enhancing survival rates [33].

Literature reviews indicate that young *Argania spinosa* plants exhibit defensive mechanisms against extreme environmental factors to ensure their survival in arid and semi-arid climates [6,8,9,28,34,35]. During periods of drought, plants tend to lower their metabolic processes [32], which subsequently enhances their survival [33]. The existing literature suggests that young plants of *Argania spinosa* exhibit defensive mechanisms against harsh environmental conditions to thrive in arid and semi-arid climates [6,8,9,28,34,35]. However, it is crucial to highlight that these studies have predominantly focused on controlled in vitro environments or field conditions. They tend to have a narrow focus, often centering on argan seeds or fully grown trees. This study seeks to fill the existing research void by examining how various environmental factors including climate and soil physicochemical properties affect the physiological and biochemical responses of young argan plants under orchard conditions.

The objective of this study is to deepen our comprehension of how young argan plants respond physiologically and biochemically to drought and salinity across three crucial areas in the Western center of Morocco. Through the examination of both climatic conditions and soil physicochemical characteristics, we aim to ascertain the tolerance levels of young argan plants in each region. We will evaluate eco-physiological metrics such as water use efficiency, photosynthesis, and growth, allowing us to effectively gauge their reactions to environmental stressors. Additionally, we will explore alterations in phytochemical compounds, integral to the medicinal and economic significance of the tree. This thorough investigation aims not only to clarify the current status of young argan plants but also to identify potential adaptation and conservation strategies. The outcomes of this research will offer valuable insights for future reforestation initiatives and guide sustainable management practices amid the challenges posed by climate change.

We propose the following hypotheses: (1) Climatic factors (such as precipitation, temperature, wind speed, and heat) vary considerably among the study sites, leading to substantial variation in the morphophysiological responses of young argan plants to these different climatic factors. (2) The soil’s physicochemical parameters influence the morphophysiological parameters of young argan plants. (3) The morphophysiological responses of young argan plants vary according to the variation in edaphic and climatic factors between sites, as well as with the variation in morphophysiological parameters between them. This results in a reduction in leaf surface, an increase in photosynthetic efficiency, and a loss of leaves in the event of severe stress.

## 2. Materials and Methods

### 2.1. Study Area

The research was carried out across six newly established argan orchards located in three regions of the Western center of Morocco. Details regarding their geographical position, altitude, climatic conditions, and soil composition are outlined in Table 1. The young argan trees involved in the study were planted in 2019. A consistent irrigation regimen was followed, with each tree receiving 40 L of water per month throughout the study duration from March 2022 to March 2023.

The study was conducted in argan orchards situated in marginal and vulnerable areas within three regions of northern Africa: Souss-Massa, Marrakech–Safi, and Guelmim-Oued Noun. These regions are all part of the Argan Biosphere Reserve, which is situated in the north of the continent (Figure 1).

### 2.2. Sampling and Field Measurement

At each site, 30 *Argania spinosa* plants were randomly chosen from the various genotypic seedling experiments planted in 2019. The measurements of aerial growth and trunk diameter of the young argan plants were conducted using a tape measure and a caliper, respectively. Taking into account the exposure factors (the four directions: north, east, south, and west), plant samples were taken with secateurs from the lower, middle, and upper parts of each plant. The leaves and branches from each plant were separated and placed into cardboard bags, which were then stored in a refrigerator at 8 °C. To evaluate the influence of seasonal climate on the physiology of *Argania spinosa*, four sampling periods were designated throughout the annual cycle of 2022/2023: March 2022, June 2022, November 2022, and March 2023.

In each orchard, four random sampling points were selected to collect soil samples at three different depths (0–20 cm, 20–40 cm, and 40–60 cm) using an auger. These soil samples were then placed in food-grade polyethylene bags. Once at the laboratory, the soil samples were stored in a cold room at 8 °C overnight before being air-dried the following day.

### 2.3. Analysis of Climatic Indices

The daily minimum, maximum, and mean air temperature values (°C); total precipitation (mm); relative air humidity (%); air temperature (°C); and wind speed (km/h), gathered from the “historique météo” site, were utilized to construct Bagnouls Gaussen’s diagrams for the period spanning March 2022 to March 2023. These diagrams illustrate the monthly averages of precipitation and temperature, alongside Tmax, Tmin, and relative air humidity, during the study period [36].

### 2.4. Laboratory Analysis

#### 2.4.1. Physiological Analysis

For the leaf surface, 150 leaves from each site were used to determine the leaf area using a leaf area meter (AM-350, ADC Bioscientific Ltd., Hoddesdon, UK), with results reported in mm^2^.

#### 2.4.2. Biochemical Analysis

The chlorophyll content was determined for 150 leaves from each site using a chlorophyll content meter (CCM-300, Opti-Sciences, Hudson, NY, USA), with the results expressed in milligrams per square meter. In order to ascertain the relative water content of the leaves and branches of each juvenile argan tree, the initial fresh weight of 150 leaves and branches was recorded for each site. Following this, the samples were subsequently oven-dried at 70 °C for a period of 24 h.

#### 2.4.3. Soil Analysis

Soil samples were air-dried and passed through a 2 mm sieve. Soil moisture was determined by calculating the difference in weight between wet and dried samples after they were oven-dried at 105 °C for 24 h [37]. Soil pH was assessed using a 1:2 soil-to-water ratio with a pH meter (C3010, Consort™, Angers, France), following Baize’s [38] classification for soil pH. Soil salinity was measured at a 1:5 ratio using an EC meter (C3010, Consort™, Angers, France), identifying saline profiles according to Aubert [39]. The limestone content was evaluated using the Bernard calcimeter method [40], and horizons were classified based on limestone content using the GEPPA method [38]. Soil organic matter (OM) was examined via the Walkley and Black method [41], while total soil nitrogen (N) was analyzed using the Kjeldahl method [42], and classified according to Calvet and Villemin [43]. Available phosphorus in the soil was estimated using the Olsen method [44], and measured via spectrophotometry (UV/VIS Spectrometer T80+ PG Instruments Limited, Leicestershire, UK). Soil fertility was evaluated by calculating the C/N ratio.

## 3. Statistical Analysis

Statistical analyses were conducted using the SPSS program (IBM Corp. Released in 2019, Version 26, New York, NY, USA). Each sampling point within each site represented a replicate for soil parameters (*n* = 12), while each young argan plant at each site also served as a replicate (*n* = 30). The data’s normality was assessed using the Kolmogorov–Smirnov test, and homogeneity of variance was tested with Levene’s test. Two-way ANOVA tests were applied, contingent upon confirming both normality and variance homogeneity, with statistical significance set at *p* = 0.05. The two-way ANOVA was used to examine the variation in morphophysiological parameters of young argan plants in relation to variation in sites, seasons, and their combinations. Similarly, this statistical technique was employed to investigate the evolution of the edaphic parameters in response to the aforementioned factors. Additionally, a principal component analysis (PCA) was performed using the basic R package under version R 4.2.2 to investigate relationships and potential similarities among various climatic conditions, physicochemical soil parameters, and the physiological traits of the young argan plants.

## 4. Results

### 4.1. Climatic Conditions in the Analyzed Orchards

This section highlights the fluctuations in precipitation, average temperatures, drought conditions, and both maximum and minimum temperatures (Tmax and Tmin) as well as relative humidity over the study period across six distinct orchards. In Figure 2, the right axis displays temperature (°C) and relative humidity (%), while the left axis indicates precipitation (mm). Notably, the scale for precipitation is set at double that of temperature (°C). The average annual temperatures varied from 22 °C in Tioughza to 27.5 °C in Imoulass, while Ezzaouite recorded the highest precipitation at 287.8 mm, contrasted with Laqsabi’s lowest at 116.9 mm.

Every area observed an extended dry season, although the length of this season differed among the sites. Rasmouka boasted the longest dry season, lasting 11 months, followed closely by Tioughza, Imoulass, and Tamjloujt, each with a 10-month dry season. The remaining location experienced a 9-month dry season throughout the study. Ezzaouite (73.8%) and Tioughza (71.2%) exhibited the highest levels of relative humidity, whereas the mountainous Imoulass had the lowest at 55.3%. Additionally, Imoulass also registered an extreme maximum temperature of 50 °C and a minimum of 5 °C, surpassing the measurements from the other sites.

Table 2 presents the fluctuations in wind speed (WS) and air temperature (AT) observed during the study period across the six chosen orchards. Ezzaouite recorded the highest wind speed at 39 km/h, while the lowest was noted in Tioughza at 12 km/h. Concerning air temperature, Imoulass reported the highest measurement at 27 °C, whereas Laqsabi, Imoulass, and Tamjloujt all recorded the lowest at 11 °C.

### 4.2. Physiological and Biochemical Parameters

#### Descriptive Statistics

Figure 3 illustrates the variations in physiological parameters of young argan plants across different study locations and throughout the four seasons. The site of Tamjloujt exhibited the most significant growth, recording height at 91.33 ± 28.69 cm and trunk diameter at 24.85 ± 8.94 mm, outperforming all other sites. In contrast, Tioughza showed the least growth in height at 52.61 ± 10.42 cm, while Imoulass had the smallest trunk diameter at 6.46 ± 1.57 mm (Figure 3A,B). During the seasons, Tioughza’s increase in aerial biomass was minimal, not surpassing 2.22%, whereas Tamjloujt achieved a remarkable growth of 21.73%. The disparity in the growth of young argan plants between these two locations is 73.59%. Additionally, the trunk growth gain in Tamjloujt exceeded 32.20% over the seasons, while Imoulass recorded a trunk growth gain of 40.04%.

The peak water content in the leaves (WCL) of young argan plants was observed in Tamjloujt in March 2022, reaching 46.89 ± 4.06%. In contrast, the lowest water content was noted in Tioughza in November 2022 at 16.44 ± 5.75%. The WCL displayed seasonal fluctuations, decreasing as temperatures rose (Figure 3C). The highest branch water content (WCB) was also documented in Tamjloujt in March 2022 at 37.76 ± 3.51% (Figure 3D), while the lowest WCB occurred in Imoulass in June 2022 at 16.80 ± 9.97%. The WCB demonstrated a similar seasonal pattern to the WCL.

The highest chlorophyll concentration was recorded in Laqsabi in November 2022, measuring 506.97 ± 92.25 mg/m^2^, whereas the minimum concentration was noted in Imoulass in March 2022 at 72.35 ± 39.18 mg/m^2^, reflecting an 85.73% reduction between the two sites (Figure 3E). The largest leaf area was found in Tamjloujt in June 2022 at 69.33 ± 19.28 mm^2^, while the smallest leaf area was measured in Tioughza in November 2022 at 29.55 ± 10.36 mm^2^, indicating a 57.38% decrease between these two locations (Figure 3F).

### 4.3. Influence of Site and Season on Physiological Parameters

Table 3 presents the mean square results from the two-way ANOVA test regarding the physiological parameters influenced by site and season. The findings reveal highly significant effects (*p* < 0.001) of both site and season on the physiological characteristics of young argan plants. Furthermore, the interaction between site and season also significantly influences (*p* < 0.001) leaf and branch water content, chlorophyll concentration, and leaf surface.

### 4.4. Soil Physicochemical Parameters

#### 4.4.1. Descriptive Statistics

The variability of soil edaphic parameters across various study sites throughout four seasons is illustrated in Figure 4. In November, Laqsabi soil exhibited mild salinity (EC = 0.79 ± 0.25 ms/cm), surpassing the non-saline threshold of 0.6 ± 0.01 ms/cm. The other study sites maintained non-saline conditions during the entire duration of the study. Laqsabi had the highest electrical conductivity (EC) at 0.79 ± 0.025 ms/cm, whereas Imoulass recorded the lowest at 0.06 ± 0.01 ms/cm (Figure 4A). The pH levels of the soil varied from moderately alkaline (pH 6.6 ± 0.21 in Imoulass) to alkaline (pH 8.6 ± 0.12 in Laqsabi) (Figure 4B).

Ezzaouite consistently reported the highest soil moisture content at 20.82 ± 1%, while Imoulass and Tamjloujt had the lowest moisture levels (2 ± 1% and 2.2 ± 0.4%, respectively). The peak soil moisture was recorded in March 2022, aligning with the Winter season (Figure 4C). The limestone (CaCO_3_) content varied from 0 ± 0% to 59 ± 7%. Ezzaouite’s soil is classified as having a high to very high limestone content (59%), while Tioughza and Imoulass demonstrate no to minimal limestone presence (0 ± 0%) (Figure 4D).

With the exception of Ezzaouite, the organic matter levels in soils from all study sites are deemed poor, ranging from 1 ± 0.21% to 2 ± 0.31%. Ezzaouite itself falls into a category of poor to moderately rich organic matter (1.7 ± 0.12%, 2.4 ± 0.45%) (Figure 4E). Laqsabi, Tioughza, and Tamjloujt soils have notably low nitrogen concentrations (N < 0.05 ± 0.01%). For the earlier sampling periods (March 2022, June 2022, and November 2022), Imoulass and Rasmouka also showed poor nitrogen levels, with a subsequent rise to low nitrogen values between 0.05% and 0.1% by March 2023. Ezzaouite generally maintained low nitrogen levels throughout the study, except for November 2022 when it was categorized as having low nitrogen levels ranging from 0.05% to 0.1% (Figure 4F).

Phosphorus availability in soils across all sites and periods was very low, fluctuating from 0 to 5 mg/kg of soil (Figure 4G). All sampled soils exhibited a C/N ratio exceeding 15, indicating a slow rate of mineralization and organic matter buildup (Figure 4H).

#### 4.4.2. Effect of Site, Season, and Soil Depth on Soil Physiochemical Parameters

The mean square of the two-way ANOVA test for the different soil physicochemical parameters across different study sites and seasons is represented in Table 4. The results indicate that both the site and season and their interaction have a highly significant effect (*p* < 0.001) on the soil’s physicochemical parameters.

### 4.5. Principal Component Analysis

Principal component analyses (PCAs) were performed to explore the relationships among edaphic, climatic, and physiological factors of young argan trees across the various study locations (Figure 5. The variability of the first two principal components, PC1 and PC2, accounted for 17.5% and 14.9%, respectively. The findings indicated a significant overlap between Tamjloujt and Rasmouka, as well as between Tioughza and Imoulass. Laqsabi and Ezzaouite also showed intersections with each other and with the other sites. Notably, there was significant convergence across the different time periods, with the exception of March 2022, which appeared to be quite distinct from the others.

The PCA revealed two primary components: Component 1 was based on factors such as precipitation, air temperature, wind speed, relative air humidity, chlorophyll, Tmax, Tmin, and soil moisture content; Component 2 comprised WCL, diameter, WCB, young plant length, pH, leaf surface, chlorophyll, and Tmin (Figure 5a,b).

Pearson’s correlation tables for the different study sites are displayed in Figure 6. These tables reveal a strong correlation between the edaphic and climatic factors and the morphophysiological characteristics of the young argan plants. Such environmental influences play a significant role in positively or negatively affecting the development of various morphophysiological traits in young argan plants. The principal component analysis (PCA), illustrated in Figure 5a, demonstrates that the environmental characteristics of Tamjloujt and Rasmouka are similar due to their proximity. There is a significant positive correlation between aboveground growth and stem diameter (R^2^ = 0.70, R^2^ = 0.63, and *p* < 0.001, respectively). A positive correlation was observed between leaf water content and leaf area, as well as branch water content (R^2^ = 0.38, R^2^ = 0.45; R^2^ = 0.80, R^2^ = 0.35; *p* < 0.01 and *p* < 0.001, respectively). Conversely, there was a negative correlation between leaf water content and soil moisture (R^2^ = 0.58, R^2^ = 0.32, *p* ≤ 0.01, and *p* ≤ 0.001, respectively). The results indicated a positive correlation between WCL and several variables, including leaf area (R^2^ = 0.38, R^2^ = 0.45, and *p* < 0.001, respectively), branch water content (R^2^ = 0.80, R^2^ = 0.35, *p* < 0.01, and *p* < 0.001, respectively), and soil moisture (R^2^ = 0.58, R^2^ = 0.32, *p* < 0.01, and *p* < 0.001, respectively). A negative correlation was identified between WCB and wind temperature (R^2^ = −0.63, R^2^ = −0.34, *p* < 0.001), as well as Tmin (R^2^ = −0.65, R^2^ = −0.44, *p* < 0.001), while a positive correlation was observed with leaf area (R^2^ = 0.38, R^2^ = 0.45, *p* < 0.001), and branch water content (R^2^ = 0.80, R^2^ = 0.35, *p* < 0.01, and *p* < 0.001, respectively). Furthermore, Tamjloujt exhibited a pronounced increase in WCB levels in association with elevated precipitation and relative air humidity (R^2^ = 0.59; R^2^ = 0.34; *p* < 0.001).

As can be observed in Figure 5a and Figure 6, there is a notable superposition between Imoulass, Tioughza, and Ezzaouite, indicating that they possess similar edaphic, climatic, and morphophysiological attributes. Furthermore, the PCA (Figure 5a) illustrates a pronounced overlap between the Tioughza and Ezzaouite samples. Ultimately, Laqsabi was found to intersect with all of the study sites.

The growth of young argan plants’ aerial parts at the study sites of Imoulass, Ezzaouite, Laqsabi, and Tioughza has been found to show significant correlations with a range of environmental factors. In Imoulass and Ezzaouite, there is a strong correlation between aboveground growth and trunk diameter (R^2^ = 0.54, R^2^ = 0.60, and *p* < 0.001, respectively). Conversely, aerial diameter at all sites is influenced by precipitation, air humidity, and Tmax (R^2^ = −0.52, R^2^ = −0.44, R^2^ = −0.38, R^2^ = −0.56; R^2^ = −0.37, R^2^ = −0.52, R^2^ = −0.34, R^2^ = −0.31; R^2^ = 0.33, R^2^ = 0.31, R^2^ = 0.50, R^2^ = 0.24; *p* < 0.001 and *p* < 0.01, respectively). In Imoulass and Ezzaouite, aboveground growth was found to have a strong relationship with trunk diameter (R^2^ = 0.54, R^2^ = 0.60, with a *p*-value less than 0.001 in both cases), while aerial diameter was shown to be influenced by precipitation, air humidity, and Tmax (R^2^ = −0.44, R^2^ = −0.38, R^2^ = −0.56; R^2^ = −0.37, R^2^ = −0.52, R^2^ = −0.34, R^2^ = −0.31; R^2^ = 0.33, R^2^ = 0.31, R^2^ = 0.50, R^2^ = 0.24, with all *p*-values less than 0.001 and some below 0.01). In Tioughza, Laqsabi, and Imoulass, trunk diameter is found to be impacted by wind speed and air temperature (R^2^ = −0.46, R^2^ = −0.50, R^2^ = −0.24; R^2^ = 0.32, R^2^ = 0.44, R^2^ = −0.47; *p* < 0.001, *p* < 0.01, and *p* < 0.05, respectively). A significant relationship is observed between leaf water content (WCL) and branch water content (WCB) across sites (R^2^ = 0.78, R^2^ = 0.76, R^2^ = 0.67, R^2^ = 0.51, and *p* < 0.001, respectively). Both WCL and WCB are influenced by various environmental variables, including air humidity, wind speed, air temperature, and soil pH. In particular, a correlation was observed between WCOL and Tmin, as well as soil moisture, in the Imoulass region (R^2^ = −0.61, R^2^ = 0.32, *p* < 0.001, and *p* < 0.01, respectively). Furthermore, chlorophyll levels are significantly influenced by factors such as wind speed, precipitation, and air humidity (R^2^ = −0.64, R^2^ = −0.52, R^2^ = −0.50, R^2^ = −0.93; R^2^ = 0.59, R^2^ = 0.63, R^2^ = 0.61, R^2^ = 0.45; R^2^ = −0.76, R^2^ = −0.63, R^2^ = −0.72, R^2^ = −0.68; and *p* < 0.001, respectively) across all sites, exhibiting a notable correlation with minimum and maximum temperatures (R^2^ = 0.63, R^2^ = 0.40, R^2^ = 0.34; R^2^ = 0.42, R^2^ = 0.24, R^2^ = 0.29; *p* < 0.001, *p* < 0.01, and *p* < 0.05, respectively), as well as soil moisture (R^2^ = −0.76, R^2^ = 0.31, R^2^ = −0.26, *p* < 0.001, *p* < 0.01, and *p* < 0.05, respectively). These results demonstrate that local climatic and soil conditions have a significant influence on the development of the argan young plants.

## 5. Discussion

This study sought to shed light on the seasonal changes in physiological parameters of young argan trees cultivated in orchards, where climatic and edaphic conditions varied across six different locations within the argan distribution area. Morocco experiences a semi-arid to arid climate, characterized by extreme temperatures, low and unpredictable rainfall, and frequent drought conditions [45]. While argan trees are recognized for their resilience to high temperatures [46], their immature stages are particularly susceptible [1]. Adequate moisture and shelter from intense sunlight and heat are crucial for young argan plants to develop healthy root systems and grow.

The principal component analysis (PCA) (Figure 5a and Figure 6) indicates a significant overlap between the areas of Tamjloujt and Rasmouka. Also, it reveals a considerable overlap among Imoulass, Tioughza, and Ezzaouite, suggesting that these areas share similar edaphic, climatic, and morphophysiological traits. Additionally, the PCA highlights a notable intersection between Tioughza and Ezzaouite. Lastly, Laqsabi intersects with all study locations.

In the aforementioned study area, the data demonstrated a significant positive correlation between growth in the aerial portion and trunk diameter for the samples from Tamjloujt, Rasmouka, Imoulass, and Ezzaouite (R^2^ = 0.70, R^2^ = 0.63, R^2^ = 0.54, R^2^ = 0.60, and *p* < 0.001, respectively). Young argan plants were found to develop both the length and diameter of their aerial structures. Ros et al. [47] noted a positive correlation between trunk diameter and growth in argan plants after one year, which they attribute to biomass accumulation. This finding is supported by numerous studies [48,49,50], which have also observed a similar trend.

The results revealed a significant negative correlation between trunk diameter growth and precipitation levels in all the study locations (Tamjloujt, Rasmouka, Imoulass, Laqsabi, Tioughza, and Ezzaouite) (R^2^ = −0.29, R^2^ = −0.31, R^2^ = −0.52, R^2^ = −0.44, R^2^ = −0.38, R^2^ = −0.56, *p* < 0.05, *p* < 0.01, and *p* < 0.001, respectively), as demonstrated by the regression coefficients. It has been postulated that mean annual rainfall may have both positive and negative effects on trunk diameter growth, depending on the species in question [51,52,53]. However, this study found no such correlation, which is likely attributable to the consistent irrigation of young argan plants. A significant correlation was identified across the five sites (Rasmouk, Imoulass, Laqsabi, Tioughza, and Ezzaouite) between trunk diameter and factors such as air humidity and Tmax (R^2^ = −0.31, R^2^ = −0.37, R^2^ = −0.52, R^2^ = −0.34, R^2^ = −0.31; R^2^ = 0.37, R^2^ = 0.33, R^2^ = 0.31, R^2^ = 0.50, R^2^ = 0.24; *p* < 0.001 and *p* < 0.01, respectively). As reported by Sánchez-Costa et al. [54], the physiological regulation of water status in a range of tree species has been shown to impact trunk diameter growth, with an observed decrease in diameter with increased relative air humidity. It was determined that factors such as relative air humidity significantly affect diameter variation [54]. Furthermore, higher temperatures (Tmax) have been found to have a positive impact on young argan trunk diameter, as evidenced by the research conducted by Auger and Fortin et al. [55,56], which observed the effect that temperature has on trunk diameter growth. In Tioughza, Laqsabi, and Imoulass, trunk diameter was significantly influenced by wind speed and air temperature (R^2^ = −0.46, R^2^ = −0.50, R^2^ = −0.24; R^2^ = 0.32, R^2^ = 0.44, R^2^ = −0.47; *p* < 0.001, *p* < 0.01, and *p* < 0.05, respectively). Previous research identified wind as a principal factor of mechanical disturbance affecting plant growth patterns [57]. The wind speed across all study areas negatively impacted young argan trunk diameter, which contrasts with most prior studies on plant species, where increased wind speed is associated with larger stem diameters [58,59]. We observed a positive impact of air temperature (AT) on trunk diameter growth in Tioughza (AT between 14 and 23 °C) and Laqsabi (AT between 11 and 23 °C), while in Imoulass (AT between 10 and 27 °C), it had detrimental effects. Young argan plants exhibited heightened sensitivity to adverse environmental factors. Prior studies have shown that lower air temperatures influence stem diameter growth rates [60,61]. In Laqsabi, trunk diameter was significantly influenced by the water content of leaves and branches (R^2^ = 0.61, R^2^ = 0.40, and *p* < 0.001, respectively). Drought conditions affect the water content within argan plant tissues, or alternatively, may impact cell division [62,63]. At Laqsabi, with a water content of 31.04% WCL and 26.78% WCB, conditions were adequate for normal cell division and a trunk diameter increase in young argan plants.

A strong correlation was observed between WCL and WCB for the following areas: Tamjloujt, Rasmouka, Laqsabi, Tioughza, Ezzaouite, and Imoulass (R^2^ = 0.80, R^2^ = 0.35, R^2^ = 0.78, R^2^ = 0.76, R^2^ = 0.67, R^2^ = 0.51, *p* < 0.01, and *p* < 0.001, respectively). There is a progressive variation in the water content of the leaves and branches of young argan plants, with transpiration exerting a direct influence on the moisture content of the leaves and stems [64,65]. Across Tamjloujt, Rasmouka, and Imoulass, WCL showed a positive correlation with soil moisture (R^2^ = 0.58, R^2^ = 0.32, R^2^ = 0.32, *p* < 0.01, and *p* < 0.001, respectively), facilitating the roots’ ability to extract water from the soil and nourish various plant compartments, as supported by other research [66,67]. However, WCL declined significantly with rising air temperatures in Tamjloujt, Rasmoka, Laqsabi, Imoulass, and Tioughza (R^2^ = −0.63, R^2^ = −0.34, R^2^ = 0.44, R^2^ = −0.55, R^2^ = −0.37, *p* < 0.01, and *p* < 0.001, respectively). As air temperatures rises, air humidity capacity increases, which in turn leads to the loss of water from plant cells via the stomata. Elevated air temperature increases the air’s hygrometric capacity, promoting water evaporation through stomata by increasing evaporation; as air temperature rises, WCL decreases due to the heightened atmospheric water demand [68]. Additionally, WCL was negatively impacted by minimum temperatures (Tmin), in Tamjloujt, Rasmouka, and Imoulass (R^2^ = −0.65, R^2^ = −0.44, R^2^ = −0.61, and *p* < 0.001, respectively). A study by Hussain et al. [69] indicated that water uptake from the soil is significantly affected by low temperatures (Tmin), resulting in a decrease in root surface area and subsequent reduction in root water absorption. This finding is corroborated by other studies, including Yan et al. and Vennam et al. [66,67]. In Tamjloujt, Laqsabi, Imoulass, and Tioughza, WCL was found to significantly increase with higher wind speeds (R^2^ = 0.63, R^2^ = −0.54, R2 = −0.34, R^2^ = 0.62, *p* < 0.01, and *p* < 0.001, respectively). Manzoni et al. [70] reported that increased wind speeds diminish leaf transpiration rates, due to elevated stomatal resistance, which helps leaves conserve water. Both WCL and relative humidity were significantly correlated in Tioughza, Laqsabi, Tamjloujt, and Ezzaouite (R^2^ = 0.70, R^2^ = −0.65, R^2^ = 0.59, R^2^ = −0.26, *p* < 0.05, and *p* < 0.001, respectively). Relative air humidity influences WCL through its effect on transpiration. Higher relative humidity corresponds to reduced atmospheric water demand, such as in Tioughza (71.2%), while Laqsabi’s aridity negatively impacted relative humidity (65.47%). A study on young silver birch (*Betula pendula*) revealed that increased air humidity induced changes in hydraulic properties [71]. The WCL in Laqsabi, Imoulass, Tioughza, and Tamjloujt was significantly correlated with precipitation (R^2^ = −0.63, R^2^ = 0.42, R^2^ = 0.42, R^2^ = 0.59, *p* < 0.001, and *p* < 0.01, respectively). In general, increased precipitation enriches the soil, facilitating root water absorption to nourish the various compartments of young plants, including their leaves [66]. Earlier studies reveal that precipitation levels and relative humidity tend to follow similar upward trends [72]. WCL was significantly correlated with soil pH, and Tmax, in Laqsabi, Imoulass, and Tioughza (R^2^ = 0.46, R^2^ = −0.38, R^2^ = −0.51; R^2^ = 0.35, R^2^ = −0.63, R^2^ = −0.49; *p* < 0.01 and *p* < 0.001, respectively). Our findings indicate that soil pH in Imoulass (ranging from 6.54 to 7.72) positively impacts WCL, whereas the pH in Tioughza (7.49 to 8.53) and Laqsabi (8.17 to 8.61) negatively affects the WCL of argan young plants. Also, as soil pH increases, WCL decreases. Additionally, Bakass et al. [73] noted that water uptake rises alongside increasing pH. In Laqsabi, we observed that as temperatures rose, water content in leaves (WCL) also increased. In contrast, in Imoulass and Tioughza, higher temperatures corresponded with decreased WCL. Young argan plants in Laqsabi have demonstrated an ability to adapt to elevated temperatures through various physiological and morphological adjustments [74]. However, in Imoulass and Tioughza, increased temperatures can impair the function and structure of macromolecules, leading to increased water loss and ultimately resulting in leaf drop and plant mortality [74,75]. In Laqsabi, WCL was significantly correlated with chlorophyll and electrical conductivity (EC) (R^2^ = 0.45, R^2^ = −0.40, and *p* < 0.001, respectively). The results also showed a pronounced progression of WCL and chlorophyll in young argan leaves. Previous studies have established that leaf water deficits affect chlorophyll synthesis and degradation [76,77]. Laqsabi recorded the highest soil electrical conductivity, which negatively impacted WCL. Research on *S. fruticosa* shoots indicated that salinity considerably alters the water content within plant tissues, with a gradual decline observed as salinity increases [78,79].

We found a substantial rise in the water content of branches (WCB) with heightened wind speed, in Tamjloujt, Tioughza, and Laqsabi (R^2^ = 0.71, R^2^ = 0.66, R^2^ = −0.25, *p* < 0.05, and *p* < 0.001). Several studies have noted that transpiration may decrease with rising wind speeds, allowing plants to conserve internal water [80,81]. Our results indicate a different response: as wind speed intensified, so did the water content of the branches, likely due to the plants’ adaptive strategies [82]. Moreover, in Tamjloujt, Imoulass, Tioughza, and Laqsabi, we observed a significant increase in WCB with rising precipitation and air humidity (R^2^ = 0.67, R^2^ = 0.54, R^2^ = 0.46, R^2^ = −0.41; R^2^ = 0.42, R^2^ = 0.30, R^2^ = 0.72, R^2^ = −0.39; *p* < 0.05 and *p* < 0.001). Abiotic factors like precipitation boost soil moisture, increasing the water available for roots, which nourishes various plant compartments [66]. Precipitation and relative air humidity exhibit similar trends over time [72], while relative air humidity influences plant water content through transpiration [71]. Conversely, in Tamjloujt, Imoulass, Tioughza, and Laqsabi, WCB significantly declined with rising air temperature (R^2^ = −0.64, R^2^ = −0.54, R^2^ = −0.41, R^2^ = 0.29, *p* < 0.05, and *p* < 0.001). Under certain climatic conditions, increased air temperature can enhance transpiration, influenced by atmospheric evaporative demand [68]. Our data indicated that in Laqsabi, as temperatures climbed, WCB also increased, whereas, in Tamjloujt, Imoulass, and Tioughza, higher temperatures led to a decrease in WCB. Additionally, in Tamjloujt, Imoulass, and Tioughza, WCB significantly rose as Tmin decreased (R^2^ = −0.55, R^2^ = −0.53, R^2^ = −0.23, *p* < 0.05, and *p* < 0.001, respectively). Hussain et al. [69] noted that lower temperatures (Tmin) reduce the surface area of stems, leaves, and roots, thereby impacting water uptake and leading to decreased water content in various plant compartments. A significant relationship was found between water content in branches (WCB) and maximum temperature, in Imoulass, Tioughza, and Laqsabi (R^2^ = −0.44, R^2^ = −0.53, R^2^ = 0.32, *p* < 0.01, and *p* < 0.001, respectively). Additionally, such temperatures heighten transpiration, diminishing the water potential and content of leaves, resulting in negative pressure that draws additional water from branches, stems, and roots via xylem transport. Consequently, branch and stem water content decreases [83,84]. EC and soil pH affect the WCB, in both the Tioughza and Laqsabi sites (R^2^ = −0.24, R^2^ = −0.47; R^2^ = −0.46, R^2^ = 0.46; *p* < 0.05 and *p* < 0.001, respectively). The results indicated that soil electrical conductivity negatively impacts WCB; in both sites, higher soil electrical conductivity was linked to the irrigation of young argan plants with brackish groundwater [85]. It has been shown that salt buildup in the root zone decreases plants’ water usage and results in less water within plant cells [86,87]. In Tioughza, soil pH (ranging from 7.49 to 8.53) negatively affected WCB, while in Laqsabi, soil pH (between 8.17 and 8.61) positively influenced WCB. Bakass et al. [73] noted that water uptake escalates with increasing solution pH.

Across all the study areas (Tamjloujt, Rasmouka, Imoulass, Tioughza, Ezzaouite, and Laqsabi), chlorophyll concentration was significantly correlated with precipitation and air temperature (R^2^ = −0.58, R^2^ = 0.43, R^2^ = −0.76, R^2^ = −0.63, R^2^ = −0.72, R^2^ = −0.68; R^2^ = 0.49, R^2^ = −0.36, R^2^ = 0.59, R^2^ = 0.63, R^2^ = 0.61, R^2^ = 0.45; and *p* < 0.001, respectively). Our findings indicated that air temperature positively influences chlorophyll concentration, contradicting earlier research suggesting that elevated air temperatures can hinder photosynthesis and degrade chlorophyll [88,89,90]. Meanwhile, precipitation may enhance the photochemical activity of chloroplasts and positively correlate with chlorophyll levels; moreover, water deficit in leaves can influence both chlorophyll synthesis and degradation [76,77,89], which, in both Tamjloujt and Rasmouka, rose significantly with the increase in leaf area. Prior research indicates a robust positive correlation between chlorophyll concentration and leaf surface, with this relationship becoming progressively stronger [91,92]. However, chlorophyll concentration declined with Tmax, in Tamjloujt, Rasmouka, Imoulass, Tioughza, and Ezzaouite (R^2^ = 0.60, R^2^ = −0.38, R^2^ = 0.24, R^2^ = 0.42, R^2^ = 0.29, *p* < 0.05, and *p* < 0.001, respectively). Temperature is also known to affect chlorophyll synthesis [93], with elevated temperatures inhibiting enzymatic reactions and potentially damaging existing chlorophyll [88]. The wind speed negatively affected the chlorophyll concentration, in Tamjloujt, Imoulass, Tioughza, Ezzaouite, and Laqsabi (R^2^ = −0.58, R^2^ = −0.50, and *p* < 0.001, respectively). It was observed that photosynthetic rates diminish at very low wind speeds due to an increase in the leaf boundary layer, which hinders carbon dioxide diffusion. However, this scenario does not apply to our study areas, where wind speeds are significantly high. In fact, a decrease in chlorophyll levels was noted in the leaves of young argan plants [94,95]. While chlorophyll concentration decreased with increased wind speed, contrary findings exist in other research suggesting that an increase in wind speed is linked to a rise in chlorophyll-a concentration [96]. Respectively, in Imoulass, Laqsabi, and Ezzaouite, significant correlations were found between chlorophyll and relative air humidity (R^2^ = 0.82, R^2^ = −0.83, R^2^ = −0.53, and *p* < 0.001, respectively). A correlation between higher humidity and reduced chlorophyll content was observed in Ezzaouite (73.8%) and Laqsabi (65.47%). Conversely, Imoulass (55.3%) demonstrated lower humidity levels. This finding suggests that elevated humidity levels are linked to reduced chlorophyll content, indicating that young argan plants adapt to low humidity by enhancing chlorophyll concentration to improve solute production and resilience in dry conditions [97,98]. A significant correlation was observed between chlorophyll levels and Tmin in Imoulass, Tioughza, and Ezzaouite (R^2^ = 0.40, R^2^ = 0.63, R^2^ = 0.34, *p* < 0.001, *p* < 0.01, and *p* < 0.05, respectively). Cold Winter temperatures can limit a plant’s photosynthetic capacity [99,100,101]. However, unlike typical scenarios, the lowest temperature noted in our study at Imoulass was 5 °C, just above freezing, which positively impacted chlorophyll levels. Many tree species can generally avoid ice nucleation to some extent by supercooling below 0 °C [102,103]. In the Ezzaouite, Tioughza, and Laqsabi areas, a significant relationship existed between soil moisture content and chlorophyll (R^2^ = −0.76, R^2^ = 0.31, R^2^ = −0.26, *p* < 0.001, *p* < 0.01, and *p* < 0.05, respectively). Ezzaouite’s limestone soil affects the plant’s water uptake [104,105], consequently influencing chlorophyll levels. In Laqsabi, increased soil salinity raises osmotic pressure, subsequently limiting water absorption in young argan plants [106,107]. A similar study on soybeans indicated that water stress impacts photosynthesis [108,109]. Our findings from Imoulass and Ezzaouite demonstrate a positive correlation between chlorophyll content in young argan plants and P_2_O_5_ (R^2^ = 0.48, R^2^ = 0.37, *p* < 0.001, and *p* < 0.01, respectively). Previous research also highlights phosphorus as a key factor in chlorophyll production [110,111,112].

The leaf area of young argan plants in Tamjloujt exhibited a significant increase with higher temperature (Tmax) (R^2^ = 0.45, and *p* < 0.001, respectively). A study on the temperature effects on the size of roses found a correlation, noting that the leaf area diminishes as temperature rises [113,114]. Conversely, leaf area significantly decreased as soil pH increased in Tamjloujt, and Laqsabi (R^2^ = −0.49, R^2^ = −0.40, and *p* < 0.001, respectively), consistent with findings by Xia et al. [115], which suggest that a soil pH range of 6.5 to 7.5 is ideal for plant growth, enabling optimal nutrient availability. In our study, the soil pH in Tamjloujt, and Rasmouka, exceeded 7.5, adversely influencing the leaf area of young argan plants. Also, a negative correlation was observed between leaf area and low temperatures in Tioughza, Laqsabi, and Imoulass (R^2^ = −0.45, R^2^ = −0.33, R^2^ = −0.28, *p* < 0.001, *p* < 0.01, and *p* < 0.05, respectively). Furthermore, earlier research has highlighted that minimum temperature is a crucial environmental factor affecting various physiognomic traits, including leaf size. This is supported by evidence showing a decrease in leaf area at lower temperatures (Tmin) [116,117,118]. The data showed a significant correlation between leaf area and Ec in both the Tioughza and Laqsabi sites (R^2^ = −0.48 and R^2^ = −0.40, respectively; *p* < 0.001 for both). It was noted that soil salinity adversely affects the leaf area of young argan plants, leading to a reduction that helps the plant conserve energy. Likewise, previous studies have indicated that *Catharanthus roseus* (L.), *Withania somnifera*, and *Salvodora persica* experience a gradual decline in leaf area with increasing salinity levels [119,120,121]. A significant correlation was observed between leaf area and air temperature in Tioughza (R^2^ = −0.46, and *p* < 0.001). An increase in air temperature correlated with a decrease in leaf area among young argan plants in Tioughza, which may serve as an adaptive strategy—reducing the leaf area exposed to air to lower evaporation rates through the stomata.

## 6. Conclusions

This study is the first to explore the influence of edaphic and climatic factors on the morphophysiological parameters of *Argania spinosa* young plants in orchards. The objective was to identify the essential environmental elements that facilitate the domestication and acclimatization of these plants. The present study demonstrated that a variety of climatic factors, including high and low temperatures, wind speed, and air temperature, as well as a range of edaphic parameters, such as pH, Ec, organic matter, and nitrogen, exert both positive and negative influences on the growth and development of young argan plants across different locations. This research also revealed the manner in which the morphophysiological attributes of young *Argania spinosa* plants influence their adaptation and acclimatization to their natural environment. The acquisition of knowledge regarding the various adaptation and survival strategies of young argan plants following transplantation will facilitate improvements in the tree’s resilience to environmental challenges and the optimization of its agricultural potential. The lack of previous research in this field allows us to propose the hypothesis that vegetation cover significantly improves nutrient uptake for young plants in their early stages. This hypothesis was demonstrated at Tamjloujt. The quality of irrigation water and soil also plays an essential role in the growth and development of these young plants. This is evidenced by the findings at Laqsabi and Tioughza, where soil EC affects the water status of young plants. It is imperative to gain an understanding of the environmental influences on *A. spinosa* if the aim is to cultivate this plant in a sustainable way and to ensure its resilience and long-term viability. Future research should concentrate on the effects of environmental factors on the biochemical parameters of young *Argania spinosa* plants.

## Figures and Tables

**Figure 1 plants-14-00126-f001:**
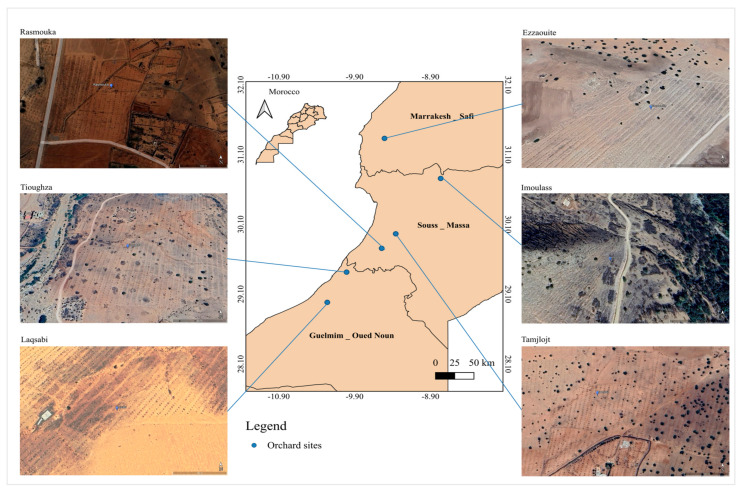
The location of the argan orchards under study in the Argan Biosphere Reserve in North Africa is indicated on the map.

**Figure 2 plants-14-00126-f002:**
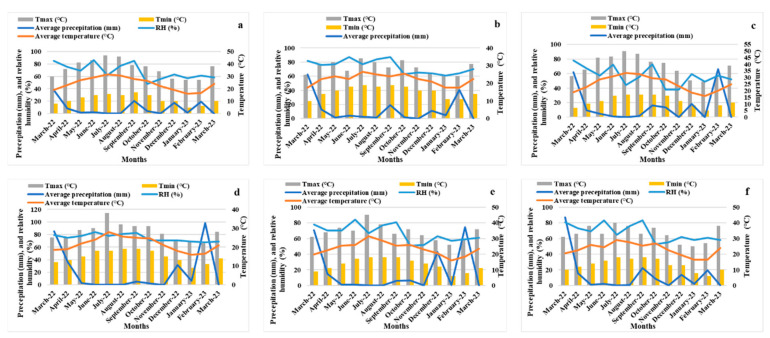
Bagnouls Gaussen’s diagrams (March 2022–March 2023) illustrate the monthly patterns of the precipitation, relative humidity (%), and maximum, minimum, and average temperatures (°C) for the sites of Laqsabi (**a**), Tioughza (**b**), Imoulass (**c**), Ezzaouite (**d**), Tamjloujt (**e**), and Rasmouka (**f**). RH denotes relative humidity (%).

**Figure 3 plants-14-00126-f003:**
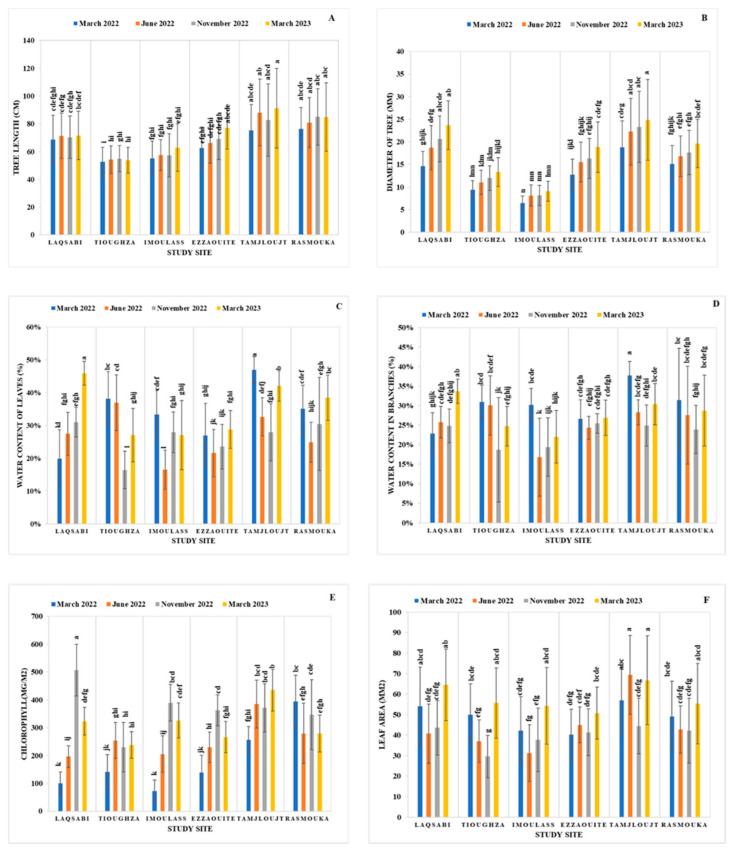
This figure displays the average values for length (**A**), diameter (**B**), leaf water content (WCL) (**C**), branch water content (WCB) (**D**), chlorophyll concentration (**E**), and leaf area (**F**) of young argan plants across various study locations throughout the four seasons. Bars sharing the same letters indicate no significant difference at the 5% significance level, based on the Tukey test. The error bars represent standard errors.

**Figure 4 plants-14-00126-f004:**
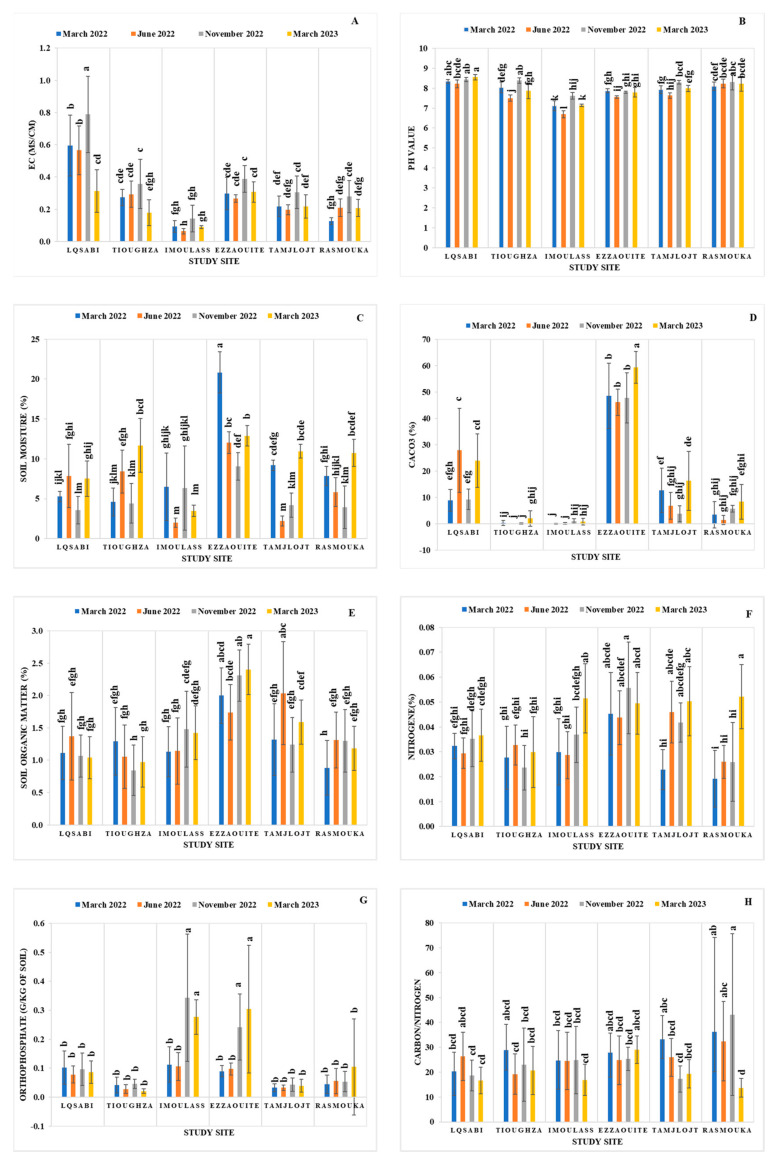
The mean of Ec: the electrical conductivity (**A**); pH: potential of hydrogen (**B**); soil moisture content (**C**); CaCO_3_ content (**D**); organic matter (OM) content (**E**); Total Kjeldahl Nitrogen (N) content (**F**); phosphorus availability, P_2_O_5_ (**G**); and the carbon/nitrogen ratio (C/N) (**H**) at the different study sites during the four seasons. The bars with the same letters are not significantly different at a 5% significance level, according to the Tukey test. Error bars refer to standard errors.

**Figure 5 plants-14-00126-f005:**
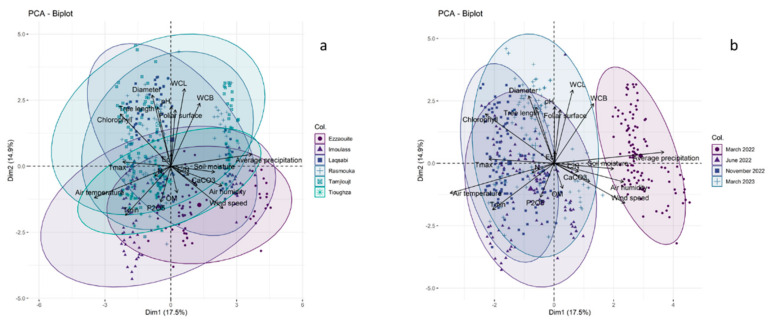
This figure showcases a principal component analysis (PCA, biplot) that highlights the differences among the six study locations, taking into account various edaphic, climatic, and physiological variables (**a**) across four seasons (March 2022; June 2022; November 2022; March 2023) (**b**). The lines radiating from the center of the biplots demonstrate both negative and positive relationships among the various variables, with their proximity indicating the strength of correlation among the physiological parameters. Key variables include pH: hydrogen potential; Ec: electrical conductivity; CaCO_3_: limestone; OM: organic matter; P_2_O_5_: available phosphorus; N: Kjeldahl nitrogen; C/N: carbon/nitrogen.

**Figure 6 plants-14-00126-f006:**
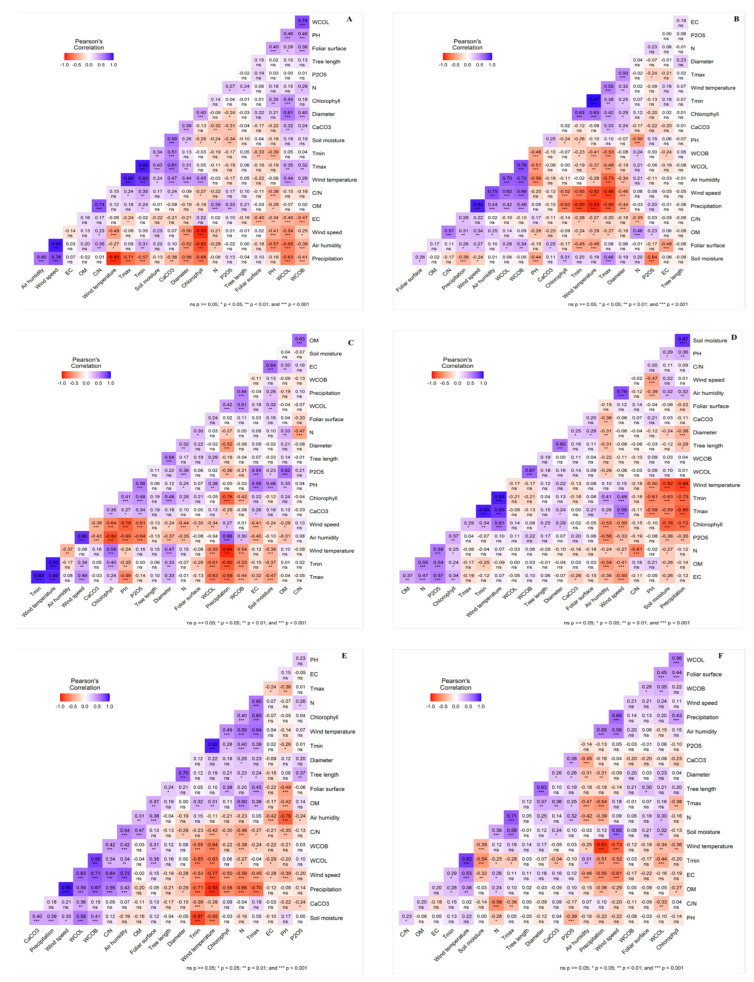
Pearson’s correlation table illustrates the variations in edaphic, climatic, and physiological parameters at Laqsabi (**A**), Tioughza (**B**), Imoulass (**C**), Ezzaouite (**D**), Tamjloujt (**E**), and Rasmouka (**F**) over a four-season period. The lines extending from the central point of the correlation cycle indicate the positive or negative associations of the different variables. Their proximity indicates the degree of correlation between the different edaphic, climatic, and physiological parameters of the young argan plants. pH: hydrogen potential; Ec: electrical conductivity; CaCO_3_: limestone; OM: organic matter; P_2_O_5_: available phosphorus; N: Kjeldahl nitrogen; C/N: carbon/nitrogen.

**Table 1 plants-14-00126-t001:** Characteristics of the different study sites.

Region	Province	Orchard Site	Geographical Localization	Altitude (m)	Climate Type	Soil Texture Classes
Guelmim-Oued Noun	Guelmim	Laqsabi	28°58′58.3″ N 10°15′34.3″ W	185	Semi-arid	Loamy sand
	Sidi Ifni	Tioughza	29°24′57.9″ N 10°00′16.6″ W	230	Subtropical desert	Silty sandy clay
Marrakech–Safi	Essaouira	Ezzaouite	31°20′01.8″ N 9°30′17.5″ W	360	Semi-arid	Sandy loam
Sous-Massa	Taroudant	Imoulass	30°45′33.2″ N 8°45′51.4″ W	1200	Temperate climate	Loamy sand
	Chtouka Ait Baha	Tamjlojt	29°57′56.4″ N 9°21′24.2″ W	239	Temperate climate	Loamy sand
	Tiznit	Rasmouka	29°45′26.1″ N 9°32′32.3″ W	192	Temperate climate	Loamy sand

**Table 2 plants-14-00126-t002:** Variations in wind speed (WS) and air temperature (AT) across the orchards for the study duration (March 2022–March 2023).

Sites	Mois Parameters	March-22	April-22	May-22	June-22	July-22	August-22	September-22	October-22	November-22	December-22	January-23	February-23	March-23
Laqsabi	WS (Km/h)	30	30	28	27	25	27	29	26	18	25	21	34	22
	AT (°C)	11	14	18	18	23	21	19	21	16	14	9	10	16
Tioughza	WS (Km/h)	25	23	16	18	14	16	18	15	12	19	17	25	14
	AT (°C)	14	17	19	20	22	21	21	23	20	19	15	14	18
Imoulass	WS (Km/h)	25	29	30	29	29	30	25	23	19	16	16	23	20
	AT (°C)	11	15	19	20	27	25	21	23	17	15	10	13	18
Ezzaouite	WS (Km/h)	31	35	30	37	36	39	35	22	25	23	30	22	31
	AT (°C)	13	15	18	19	22	21	21	21	18	18	13	13	16
Tamjloujt	WS (Km/h)	21	22	20	18	19	18	18	17	16	16	17	23	17
	AT (°C)	12	16	19	20	24	22	21	23	18	15	11	11	17
Rasmouka	WS (Km/h)	24	22	20	20	17	18	19	18	15	17	17	34	22
	AT (°C)	12	15	18	19	23	21	20	22	18	16	12	10	16

**Table 3 plants-14-00126-t003:** Mean Square from the two-way ANOVA analyzing the impacts of Site, Season, and their interaction on the physiology of *Argania spinosa* young plants during the study period from March 2022 to March 2023.

Factors	Tree Length (cm)	Diameter of Tree (mm)	Water Content of Leaves (%)	Water Content of Branches (%)	Chlorophyll (mg/m^2^)	Foliar Surface (mm^2^)
Site	17,891.36 ***	3323.42 ***	2206.35 ***	861.56 ***	339,668.51 ***	4934.16 ***
Season	2158.583 ***	899.45 ***	3379.55 ***	1653.80 ***	1,051,902.16 ***	9920.88 ***
Site × Season	271.66 ^ns^	28.62 ^ns^	1556.92 ***	441.16 ***	194,316.02 ***	1040.84 ***

Mean square in each column followed by *p*-value; ns: not significant; ***: very highly significant, *p* < 0.001.

**Table 4 plants-14-00126-t004:** Two-way ANOVA of the effects of site, season, soil depth, and their interactions on the edaphic parameters of young *Argania spinosa* plants during the study period.

Factors	Site	Season	Site × Season
Ec	1.785 ***	0.481 ***	0.096 ***
pH	13.552 ***	4.535 ***	0.525 ***
Soil moisture	720.347 ***	462.726 ***	165.265 ***
CaCO_3_	25,978.303 ***	1087.140 ***	412.578 ***
OM	11.314 ***	0.540 ^ns^	1.162 ***
N	0.004 ***	0.004 ***	0.001 ***
P_2_O_5_	0.410 ***	0.174 ***	0.061 ***
C/N	1068.621 ***	1588.546 ***	635.016 ***

Mean square in each column followed by *p*-value; ns: not significant; ***: very highly significant, *p* < 0.001. EC: electrical conductivity; OM: organic matter; N: Kjeldahl nitrogen; P_2_O_5_: available phosphorus; C/N: carbon/nitrogen ratio.

## Data Availability

The original contributions presented in the study are included in the article, further inquiries can be directed to the corresponding authors.

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
