# Peer review of "The Influence of Edaphic and Climatic Factors on the Morphophysiological Behavior of Young Argan Plants Cultivated in Orchards: A Comparative Analysis of Three Regions in Southwest Morocco"

_plants, 2025, doi:10.3390/plants14010126_

Round 1

Reviewer 1 Report

Comments and Suggestions for Authors

The manuscript investigated the impact of various climatic and edaphic factors on the morphological and physiological traits of young Argania spinose plants across six distinct orchards and over four seasons. It is an interesting topic for plant ecology focusing on the interactions between young argan and soil, climatic factors. However, the current version of the study is still lack of enough information for the research. I have some suggestions for the manuscript as following.

1.       In the Abstract, the findings or results did not fully reflect all the important finding of the research. The authors just offered some specific data of their results.

2.       In the last paragraph of the Introduction, I suggest the authors should add their hypothesis of the research, so that the readers can understand the key issues. Also, the hypothesis should be more specific according to their objective.

3.       In Table 1, I suggest the authors should add a map including all their studied sites, would be better than the table presentation.

4.       In Figure 2 and Figure 3, I wonder if there exist a better way for presenting the significant difference among all bars consisting all studied sites and season? May be it should present the significant difference among different season within the same sites or the significant difference among different sites within the same seasons?

5.       The data in Figure 5 is too small to be seen clearly, please revised them.

6.       The discussion is too long, I suggest the authors should rewrite according to their hypothesis. Also, it should be more concentrated that can reflect their findings.

Comments on the Quality of English Language

The English could be improved to more clearly express the research.

Author Response

Thank you very much for taking the time to review this manuscript. Please find the detailed responses below and the corrections highlighted in the re-submitted files.

Reviewer 2 Report

Comments and Suggestions for Authors

In this study, authors evaluated the influence of edaphic and climatic factors on the morphophysiological behavior of young argan plants cultivated in orchards. They tested a lot of plant and soil variables and find some rules. After carefully reading the paper, I think the manuscript should be improved before publication. The major comments are as follows.

I think the three hypotheses should be improved, as they can not be supported by the data in the manuscript. The first hypothesis referred as that “climatic factors (such as precipitation, temperature, wind speed, and heat) vary significantly from one location to another within the range of Argania spinosa”. As we known that if two locations are close, their climatic conditions will be similar, but if two locations are far apart, their climatic conditions will be very different. It's not a hypothesis, it's common sense. Moreover, six newly established argan orchards were selected in the manuscript. Are these six sites represented all the Argania spinosa region? If not, the first hypothesis cannot be supported.

The second hypothesis referred that “The irrigation of young argan plants considerably alters soil physicochemical characteristics, marked by increased electrical conductivity and seasonal fluctuations in soil fertility”. The experimental design in the manuscript has nothing to do with irrigation.

More information about the six sites should be provided. For example, are the thirty chosen young Argania spinosa plants are in the same age, same size and from the same genotype? In each location, are the density of Argania spinosa similar? Are these six locations had the similar size? How did authors select leaves and branches samples of each plant? How did the soil samples were stored after sampling?

More information about the statistical analysis should be provided. Specify the fixed factor about the two-way and three-way ANOVA tests.

In Tables 3 and 4, I don’t think using season as one of the fixed factors is rational (March 2022, June 2022, November 2022, and March 2023), as in March samples were collected twice.  

Statistical results showed that soil depth had significant effects on soil physicochemical parameters (Table 4), however, how did soil depth affect soil physicochemical parameters was not illustrated as other factors.     

Line 337-341, the important correlations among variables were not fully illustrated in Results.

The discussion should be shortened. It is too long.

Author Response

(The authors gave the same response as above.)

Reviewer 3 Report

Comments and Suggestions for Authors

The manuscript aims to investigate the influence of various edaphic and climatic factors on the morphophysiological behavior of argan plants in Southwest Morocco. The manuscript present theoretical and practical importance for argan technology. The study is generally well structured. The results are interpreted statistically and represented by clear graphs. Authors emphasized the role of findings for rehabilitation and sustainability in argan agroecosystems. The study is a multi-year one.

There are some shortcomings:

-        Mention for all equipment’s and soft programs at least the type, company and country of origin.

-        In introduction section should include more specific scientific literature for argan technology.

-        Includes more details related to the methods for: Physiological Analysis, Biochemical Analysis,

-        Did the study consider only one variety? If yes, describe. 

Author Response

(The authors gave the same response as above.)

Round 2

Reviewer 1 Report

Comments and Suggestions for Authors

I think the authors have revised the manuscript carefully and improve the quality of the paper, thus it could be considered for publication in the journal.

Reviewer 2 Report

Comments and Suggestions for Authors

The author has made corresponding revisions based on my suggestions. I think the work is ready for publication.

Reviewer 3 Report

Comments and Suggestions for Authors

The authors generally responded well to the comments and suggestions made in the review. The quality of the manuscript has been improved. The article presents theoretical and practical importance.